# Coexistence of Native and Invasive Freshwater Turtles: The Llobregat Delta (NE Iberian Peninsula) as a Case Study

Marc Franch [1,2,*], Gustavo A. Llorente [3], Maria Rieradevall [4,†], Albert Montori [5] and Miguel Cañedo-Argüelles [4,6]

1   CICGE—Centro de Investigação em Ciências Geo-Espaciais, Observatório Astronómico Prof. Manuel de Barros, University of Porto, 4430-146 Vila Nova de Gaia, Portugal
2   Biologia Animal Research Group, Departament de Ciències Ambientals, University of Girona, 17003 Girona, Spain
3   Section of Zoology and Anthropology, Department of Evolutionary Biology, Ecology and Environmental Sciences, Faculty of Biology, University of Barcelona, 08028 Barcelona, Spain
4   FEHM-Lab, Department of Evolutionary Biology, Ecology and Environmental Sciences, University of Barcelona, 08028 Barcelona, Spain
5   Centre de Recerca i Estudis Ambientals de Calafell (CREAC/GRENP), 43882 Calafell, Spain
6   Institute of Environmental Assessment and Water Research (IDAEA-CSIC), 08034 Barcelona, Spain
*   Correspondence: marc.franch@udg.edu
†   In Memoriam.

**Abstract:** The global degradation of wetlands is increasing their susceptibility to invasions, which is greatly determined by a niche overlap between native and invasive species. We analyze its role in regulating the coexistence of the native Mediterranean stripe-necked terrapin *Mauremys leprosa* and the invasive Red-eared Slider *Trachemys scripta elegans* in a coastal wetland. We analyzed both water chemistry and landscape attributes, using variance-partitioning analysis to isolate the variance explained by each set of variables. Then, the influence of environmental variables on species co-occurrence patterns was assessed by using latent variable models (LVM), which account for correlation between species that may be attributable to biotic interactions or missing environmental covariates. The species showed a very low niche overlap, with clear differences in their response to environmental and landscape filters. The distribution of *T. s. elegans* was largely explained by landscape variables, preferring uniform landscapes within the daily movement buffer, whereas at larger scales, it was associated with a high diversity of habitats of small and uniform relative sizes. A high percentage of the distribution of *M. leprosa* was unexplained by the measured variables and may be related to the competitive exclusion processes with *T. s. elegans*. The species was positively related with large patches with high perimeter values or ecotone area at medium spatial scales, and it was benefited from a marked heterogeneity in the patches' size at larger scale. According to latent variable models, both species had wide eutrophication and salinity tolerance ranges, but they showed different environmental preferences. *T. s. elegans* was related to eutrophic freshwater environments, whereas *M. leprosa* was related to more saline and less eutrophic waters. Our results suggest that *M. leprosa* modifies its habitat use in order to avoid interaction with the *T. s. elegans*. Thus, management actions aimed at removing the invasive species from the territory and promoting habitat heterogeneity might be needed to protect *M. leprosa* and avoid local extinctions.

**Keywords:** landscape structure; latent variable models; invasive species; niche selection; coexistence; reptiles; *Mauremys leprosa*; *Trachemys scripta*; biodiversity conservation

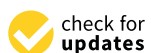



## 1. Introduction

The introduction of exotic species is currently one of the main threats to global biodiversity [1–4]. Most studies on invasive species have focused on the ecological traits of the species or their competitive ability (e.g., [5,6]). However, some authors have suggested

that niche width is a key factor influencing invasion success (e.g., [7–9]) and its impact on native communities [10–13]. The concept of ecological niche has been defined in various ways (e.g., [14–17]). The "niche breadth–invasion success" hypothesis represents the first attempt at the generalization that species have attributes that make them successful invaders [18]. It suggests that species with broad niches (generalists) are more likely to invade new regions than species with narrow niches (specialists) [18–20]. Geographic and climatic niche have been used to determinate the invasiveness of introduced species [18,21,22], but fine-grained studies are needed to understand the mechanisms and consequences of species introductions [12,23,24].

Modification of the landscape by humans has undoubtedly been a key factor in the introduction of foreign species [25–27]. Anthropogenic alteration of ecosystems is promoting invasion success because the spread of invasive species occurs more rapidly in fragmented landscapes [28–30], and habitat destruction favors invasions by habitat generalists [31–34]. Within this context, wetlands are interesting ecosystems in which to study the dynamics of invasions, since they are one of the most degraded and, at the same time, most biodiverse ecosystems of the world [35–39].

Semi-aquatic organisms (e.g., insects, amphibians, reptiles) that depend on aquatic and terrestrial habitats to complete their life-cycle and maintain viable populations are threatened by the degradation of both wetlands and their associated terrestrial habitats [40–43]. Despite the global decline in populations of many freshwater turtle species, the response of this group to habitat fragmentation has been poorly described [44–47]. Most studies on the ecological niche of freshwater turtles have focused on segregation at the microhabitat level and on feeding strategies [48–51]. However, how does a native turtle species such as *Mauremys leprosa* respond to a highly invasive and competitive species such as *Trachemys scripta elegans* at larger spatial scales and in multiple niche dimensions in human-altered environments? Existing evidence suggests that niche overlap is likely to be important for answering this question [52–55]. The main goal of this study was to analyze the factors determining the coexistence of *M. leprosa* and *T. s. elegans* in a coastal wetland heavily modified by human activity (Llobregat Delta, Spain). Specifically, we aimed to (1) determine the extent of co-occurrence between the two species and (2) quantify the role of environmental and landscape variables for coexistence of these two species.

## 2. Materials and Methods

### 2.1. Study Site and Species Description

The Llobregat Delta plain is formed by the Llobregat River estuary, lakes, marshes and flood-zone grasslands, irrigation channels, agricultural, urban and industrial areas, dunes, coastal pine forests, and major infrastructure development (i.e., Barcelona's airport and port) (Figure 1). Artificial habitats and agricultural fields occupy about 95% of the delta surface (Table 1). Due to its geomorphology and its fluvial origin, the Llobregat Delta is especially rich in aquatic environments and provides an important habitat for freshwater turtles. This area has an important population of autochthonous *Mauremys leprosa* [56] and at the same time, exotic freshwater turtles (*Trachemys scripta elegans*) are often observed [57,58] at very high densities [56,59].

The Mediterranean pond turtle (*M. leprosa*), is mainly distributed in countries surrounding the Mediterranean Sea (mainly Tunisia, Algeria, Morocco, Spain, Portugal and in south-western France) [60]. *Mauremys leprosa* is a thermophilic freshwater species and is not very selective in aquatic habitat requirements [61,62]. Its diet appears both opportunistic and omnivorous [60]. The species is classified as "Vulnerable" in the European Red List of Reptiles and in the Spanish Red List [63,64].

The red-eared slider (*T. s. elegans*) is a subspecies native to the south-western United States. Its native range extends from Virginia to north-eastern Mexico, occupying practically the entire Mississippi basin [65,66]. In this original distribution, *T. s. elegans* is considered a habitat generalist, being present in a wide variety of continental aquatic environments characterized by soft bottoms, minimal or no current and abundant vegetation [67–69]. The



species is considered omnivorous with a wide spectrum of food resources, both animal and plant, and with a clear tendency toward carnivory in newborns and juveniles and vegetarianism in adults [68,70].

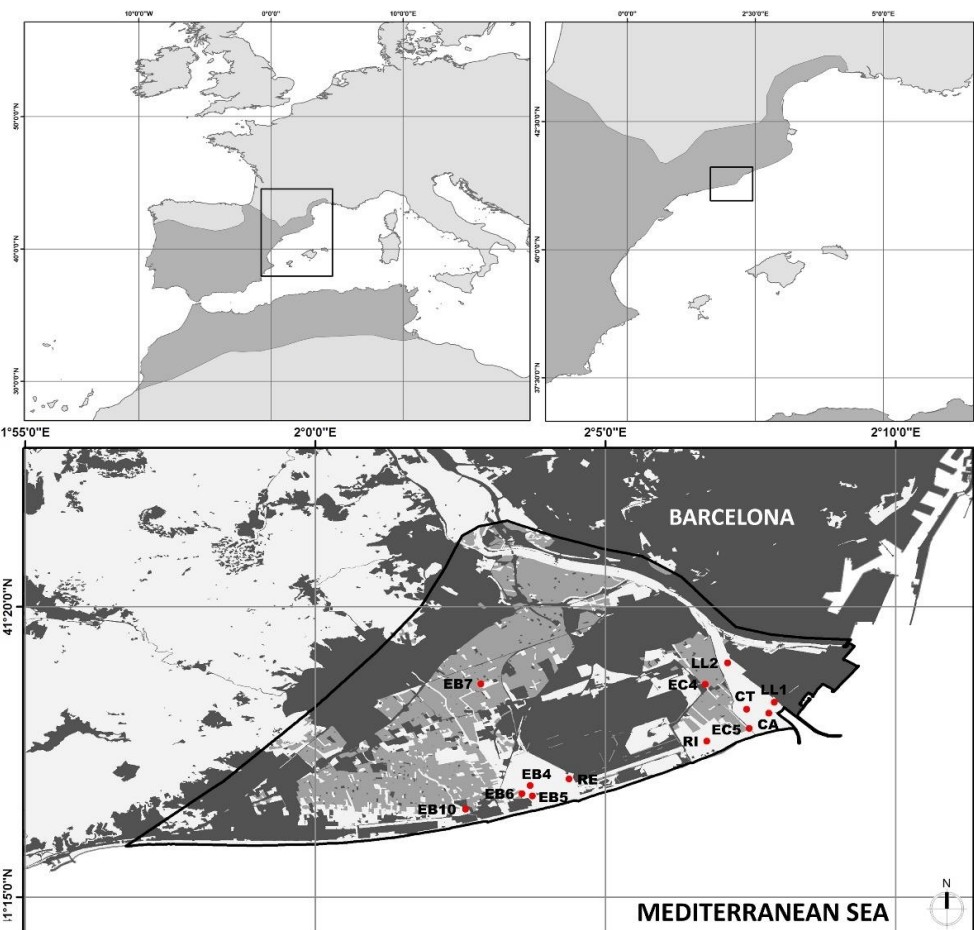

**Figure 1.** Location of the study area (Llobregat Delta, Spain) in the distribution range of *Mauremys leprosa* in the Mediterranean coast of the Iberian Peninsula (squares on top). Mainland-use categories of study area: dark gray: unproductive artificial; gray: semi-natural (essentially fields); pale gray: natural (wooded areas, wetlands, water bodies, etc.). The red dots represent our sampling stations: Ca l'Arana (CA), Cal Tet (CT), la Murtra (EB10), Bassa dels Pollancres (EB4), Braç de la Vidala (EB5), Riera de Sant Climent (EB6), Can Dimoni Gran (EB7), canal de la Bunyola 2 (EC4), canal de la Bunyola 1 (EC5), Llera Nova 1 (LL1), Llera Nova 2 (LL2), el Remolar (RE) and la Ricarda (RI).

**Table 1.** Mainland-use categories and their occupied surface within the study area (Llobregat Delta, Spain). Natural landscape does not exceed 7% of total surface.

| Category | | Surface (km$^2$) | Percentage |
|---|---|---|---|
| Artificial | Unproductive artificial | 51.31 | 78.64 |
| Seminatural | Fields | 9.81 | 15.03 |
| Natural | Dense wooded | 2.31 | 3.54 |
| | Shrublands | 0.85 | 1.30 |
| | Unproductive natural | 0.45 | 0.68 |
| | Continental waters | 0.41 | 0.62 |
| | Wetlands | 0.06 | 0.09 |
| | Light wooded | 0.03 | 0.05 |
| | Meadows and grasslands | 0.03 | 0.04 |
| | TOTAL | 65.24 | 100.00 |

*T. s. elegans* is the most widespread alien turtle in the world [71–73]. It is currently considered one of the 100 most dangerous invasive species worldwide [74]. The invasive species causes predation [75–77], competition [78–80], hybridization [81–83] and disease transmission [84–86] against native species, with the consequent loss of biodiversity of native ecosystems [87–90].

Multiple studies highlight evidence of conflict between *T. s. elegans*, the alien species, and *M. leprosa*, the native species (e.g., [91,92]). *M. leprosa* avoids interaction with *T. scripta* [93], the alien species competes efficiently for basking areas and food under experimental and natural conditions [94–97] and can transmit diseases and parasites to the native species [98–100].

## 2.2. Sampling Methodology

Thirteen water bodies with different characteristics were sampled twice per month (Figure 1) from February to November 2004 and from February to November 2005. In each sampling occasion, we recorded (1) geographic coordinates of turtle collection location and different associated variables such as the number of captures, traps, time, and (2) habitat features, measured at the local and the landscape levels. Turtles were captured using nets and baited funnel traps [101,102]. The traps were installed in different locations within each water body, making sure that they were close to water chemistry sampling stations. Twenty-four hours after installation, traps were inspected and all captured turtles marked by making marginal cuts on the carapace scutes following an international procedure for capture–mark–recapture of turtles and measured [102]. Immediately, individuals were released at the site of capture, except for the last sampling campaign, in which all the collected *T. s. elegans* were transferred to a wildlife rehabilitation center (Centre de Recuperació d'Amfibis i Rèptils de Catalunya—CRARC) for its management as an invasive alien species. During the study period, 863 freshwater turtle individuals were captured. All capture points were integrated into a Geographic Information System, ArcGIS 10.2 [103] and QGis 2.4.0-Chugiak [104]. Only the first captures of each individual (n = 374: 230 *M. leprosa* and 144 *T. s. elegans*) of total caches were analyzed, ignoring re-captured individuals in order to avoid any intraspecific or interspecific biases.

## 2.3. Landscape and Environmental Variables

For each first turtle location, three different buffers were generated: 100 m in diameter to include proximity movements (daily movements), 500 m for movements related to the annual cycle of activity and 2000 m for movements that occur occasionally (dispersive movements related to a change in the environmental conditions, in demography, etc.) [105–108].

Landscape structure was used to explore how the landscape affects the distribution of freshwater turtles, calculating different landscape parameters obtained from land-use cartography (Catalonian Land Cover Cartography [109]) for each individual. Eighteen landscape variables for each buffer were computed using the Patch Analyst Tool [110,111] implemented in ArcGIS 10.2. The variables considered were: three measures of patch richness, diversity and evenness; five patch shape and fractal dimension metrics; four edge density metrics, four patch size metrics; and two landscape descriptive metrics (Table 2). Once all variables had been registered for each captured individual, the mean value for each species per buffer combination (i.e., 100, 500 and 2000 m in diameter) at each sampling station was estimated (Supplementary Materials, Table S1). Landscape variables changed minimally during the study period (2 years).

Different variables related to water chemistry were taken (Table 2). Conductivity, pH, water temperature, and dissolved oxygen were measured using a multiparametric sensor (WTW, multiparameter model 197i), and water transparency was measured through Secchi disk depth. A surface-water sample (1.5 l) was collected at each site and preserved at 4 °C for laboratory analysis of nutrients ($NH_4^+$, $NO_3^-$, $NO_2^-$, $PO_4^{3-}$, TP and $SiO_4^{2-}$), total organic carbon (TOC), suspended solids (SSP), major ions ($SO_2^{4-}$, $Cl^-$, $Ca^{2+}$, $Mg^{2+}$, $Na^+$, $K^+$) and phytoplanktonic chlorophyll-*a* (chl-*a*) following standard methods [112] (Supplementary Materials, Table S2).

**Table 2.** Landscape descriptive metrics computed from a land-use cartographic database [109] and environmental variables considered in analyses.

| Landscape Descriptive Metrics | Metric | Description |
|---|---|---|
| Patch richness, diversity and evenness | R | Richness |
| | SDI | Shannon's diversity index |
| | SEI | Shannon's evenness index |
| Patch shape and fractal dimension | AWMSI | Area weighted mean shape index |
| | MSI | Mean shape index |
| | MPAR | Mean perimeter–area ratio |
| | MPFD | Mean patch fractal dimension |
| | AWMPFD | Area weighted mean patch fractal dimension |
| Edge density | TE | Total edge (m) |
| | ED | Edge density |
| | MPE | Mean Patch Edge (m) |
| | PSCoV | Patch size coefficient of variance |
| Patch size | MedPS | Median patch size (m$^2$) |
| | MPS | Mean patch size (m$^2$) |
| | NumP | Number of patches |
| | PSSD | Patch size standard deviation (m$^2$) |
| Landscape descriptive variables | CA | Total core area (m$^2$) |
| | TLA | Landscape area (m$^2$) |
| **Environmental Variables** | **Metric** | **Description** |
| General variables | Ox | Dissolved oxygen in water (mg/L) |
| | Ox% | Dissolved oxygen saturation in water (%) |
| | pH | pH of water |
| | Secchi | Water transparency (m) |
| | T | Temperature (°C) |
| Primary production and nutrient concentration | Chl-a | phytoplanktonic chlorophyll-*a* concentration (µg/L) |
| | DIN | Dissolved inorganic nitrogen concentration (mg/L) |
| | $NH_4^+$ | Ammonium concentration (mg/L) |
| | $NO_2^-$ | Nitrite concentration (mg/L) |
| | $NO_3^-$ | Nitrate concentration (mg/L) |
| | $PO_4^{3-}$ | Phosphate concentration (mg/L) |
| | $SiO_4^{2-}$ | Silicate concentration (mg/L) |
| | SRP | Soluble reactive phosphorous concentration (mg/L) |
| | TOC | Total organic carbon concentration (mg/L) |
| | TP | Total phosphorous concentration (mg/L) |
| Conductivity and ion concentration | $Ca^{2+}$ | Calcium concentration (mg/L) |
| | $Cl^-$ | Chloride concentration (mg/L) |
| | Cond | Water conductivity (µS/cm) |
| | $Fe^{2+}$ | Iron concentration (mg/L) |
| | $K^+$ | Potassium concentration (mg/L) |
| | $Mg^{2+}$ | Magnesium concentration (mg/L) |
| | $Mn^{2+}$ | Manganese concentration (mg/L) |
| | $Na^+$ | Sodium concentration (mg/L) |
| | $Si^{2+}$ | Silicon concentration (mg/L) |
| | $SO_2^{4-}$ | Sulphates concentration (mg/L) |
| | SSP | Suspended solids concentration (mg/L) |

*2.4. Data Analysis*

Co-occurrence between the two species was calculated from the community matrix (presence/absence data of the two species in our study sites) using the Schoener index [113] in the function niche.overlap (R package spa [114]). This function allows for using "species lists" (lists of species generated from short-term ecological censuses within areas of relatively homogeneous habitat) to compute species co-occurrence based on null model

algorithms [115]. Later, the influence of environmental and landscape variables on the co-occurrence of species was examined through variance partitioning analysis. Initially, to reduce multicollinearity from multivariate analysis, each landscape variable with the lower biological meaning from any pair of variables having a Spearman correlation coefficient higher than 0.70 or lower than −0.70 [116,117] was removed. Then, the *varpart* function in the vegan package [118] was used in order to isolate the variance in species occurrence (i.e., number of captures of each species at each site) explained by each set of abiotic variables (i.e., environmental and landscape variables) and their combined effects. The partitioning is based on redundancy analysis (RDA), and the function uses adjusted $R^2$ to assess the partitions explained by the explanatory variables and their combinations [119]. After that, different analyses were applied to environmental and landscape variables. This is because these two sets of variables had different properties. Environmental variables were measured once at each location and represented the environmental characteristics of the place in which species were captured (therefore being equal for both species), whereas landscape variables were calculated based on a buffer around the precise place in which each individual was captured and, therefore, were different for each species.

RDA analysis was used to explore the relationship between environmental variables and the abundance of each species using the function *cca* in the vegan package [118]. Then, the influence of environmental variables on species co-occurrence patterns was assessed through latent variable models (LVM [120]), which can be regarded as an extension of factor analysis [121], following Letten et al. [122]. LVMs use latent variables as a parsimonious means of modeling residual species correlation [123], which accounts for any residual correlation between species not attributable to spatial heterogeneity in the measured environmental variables. This correlation may be driven by biotic interactions such as competition (negative) or facilitation (positive) or alternatively to missing predictors. After fitting the LVMs, in order to visualize patterns of co-occurrence arising from the different environmental factors, two types of correlation matrices were calculated. The first was constructed by calculating the correlation between the fitted values of the two species [122], representing the correlation between species that can be attributed to a shared/diverging environmental response. The second type of correlation matrix was calculated using the latent variable coefficients, also known as factor loadings. This second residual correlation matrix represents the correlation between species that may be attributable to biotic interactions or missing environmental covariates. Since Bayesian MCMC estimation was used, the correlation between fitted responses was calculated for each MCMC sample, which made it possible to obtain a posterior distribution for each cell of the environmental and residual correlation matrix. As such, correlation "significance" was evaluated on the basis of the 95% credible intervals for the posterior mean excluding zero. Bayesian MCMC was performed through JAGS v3.4.0 [124] using the package R2jags v0.03-08 [125]. For each species, the most relevant landscape variable explaining the species occurrence and abundance were selected using a stepwise algorithm (function "step" in R "stats" package). Multiple generalized linear-regression models (including all possible combinations of landscape variables) were fitted, and the model with the lowest Akaike's information criterion was selected (AIC [126,127]). All statistical analyses were carried out using the statistical computing software R 3.5.0 [128].

## 3. Results

We captured 374 individuals, 230 corresponding to *M. leprosa* and 144 to *T. s. elegans* (Table 3). The two species showed a co-occurrence of 46.15% in the studied water bodies, 38.46% of the water bodies only had *T. s. elegans* and 15.38% with only *M. leprosa* (Table 3, Figure 2). After multicollinearity analysis, nine environmental variables (pH, T°, DIN, SRP, SSP, TOC, $SO_2^{4-}$ and $K^+$) and 15 landscape variables (SEI(Ø100), NumP(Ø100), PSCoV(Ø100), PSSD(Ø100), CA(Ø100), MSI(Ø500), MPAR(Ø500), CA(Ø500), R(Ø2000), SEI(Ø2000), MPAR(Ø2000), ED(Ø2000), MedPS(Ø2000), PSCoV(Ø2000) and CA(Ø2000)) were retained. The distribution of *T. s. elegans* was largely explained by landscape variables

(39.13% of total variance), whereas *M. leprosa* showed a very high percentage (75.60%) of unexplained variance (Figure 3).

**Table 3.** Sampling stations and number of captures per species.

| Station Name | Code | Longitude (E) | Latitude (N) | Typology | Captures | |
|---|---|---|---|---|---|---|
| | | | | | *M. leprosa* | *T. s. elegans* |
| Braç de la Vidala | EB5 | 2.0605 | 41.2857 | Irrigation channel | 0 | 17 |
| Canal de la Bunyola 1 | EC5 | 2.1243 | 41.2987 | Irrigation channel | 0 | 1 |
| Canal de la Bunyola 2 | EC4 | 2.1150 | 41.3076 | Irrigation channel | 35 | 85 |
| Llera Nova 1 | LL1 | 2.1307 | 41.3061 | Estuary | 1 | 0 |
| Llera Nova 2 | LL2 | 2.1169 | 41.3189 | Estuary | 0 | 2 |
| Cal Tet | CT | 2.1221 | 41.3056 | Lagoon | 68 | 7 |
| La Murtra | EB10 | 2.0396 | 41.2772 | Lagoon | 0 | 9 |
| El Remolar | RE | 2.0723 | 41.2817 | Lagoon | 9 | 2 |
| La Ricarda | RI | 2.1151 | 41.2927 | Lagoon | 49 | 3 |
| Riera de Sant Climent | EB6 | 2.0660 | 41.2771 | Lagoon | 0 | 1 |
| Ca l'Arana | CA | 2.1300 | 41.3037 | Lagoon | 47 | 0 |
| Can Dimoni Gran | EB7 | 2.0480 | 41.3110 | Pond | 1 | 12 |
| Bassa dels Pollancres | EB4 | 2.0655 | 41.2813 | Pond | 20 | 5 |
| | | | **Total:** | | **230** | **144** |

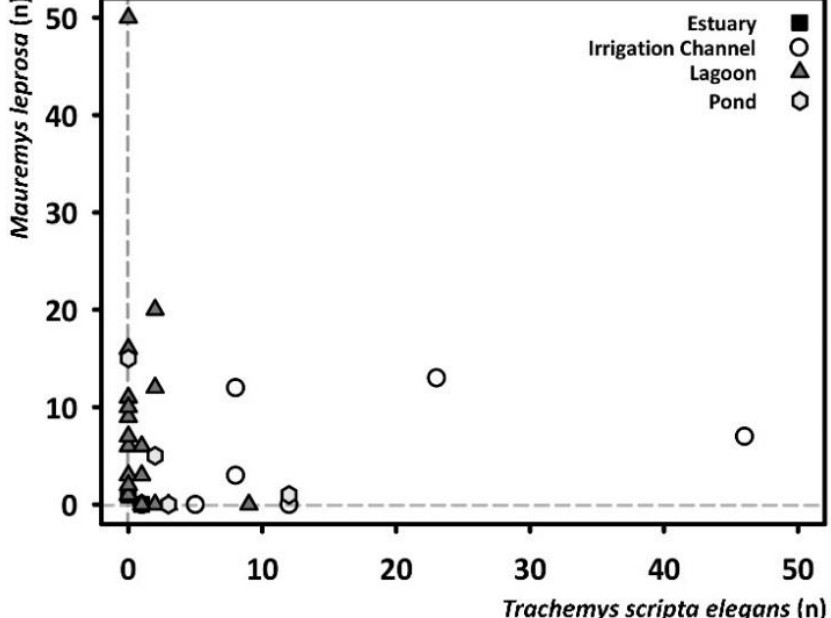

**Figure 2.** Abundance of each species (*M. leprosa* and *T. s. elegans*) at each sampling site in relation to each other. Zero values for each species are marked with a short gray dash line.

Two main environmental gradients were identified by the RDA analysis: nutrient enrichment (i.e., concentrations of SRP and the different forms of nitrogen) and salinity (i.e., conductivity and ion concentrations). Although there was no clear differentiation of each species along the two environmental gradients, *T. s. elegans* tended to dominate in areas with higher nutrient concentrations and lower salinity than *M. leprosa* (Figure 4). Despite this, both species seemed to prefer sites with low nutrient enrichment and salinity (Figure 4).

LVM yielded negative correlations (i.e., due to divergence in the environmental preferences of the species) for all environmental variables except for the $Na^+$ (Table 4), although these were weak and not significant. Only SRP, ammonia ($NH_4$), chlorophyll-*a* (Chl-*a*) and suspended solids (SSP) seemed to have some importance in explaining the co-occurrence

of both species (Table 4). *M. leprosa* preferred lower ammonium, chlorophyll-*a* and phosphorous concentrations, and it tolerated higher SSP concentrations (highly correlated with conductivity) than *T. s. elegans* (Figure 5). The mean residual correlation was −0.48, meaning that 48% of the variation in the co-occurrence of the two species was explained by their biotic interaction or by variables that we did not measure.

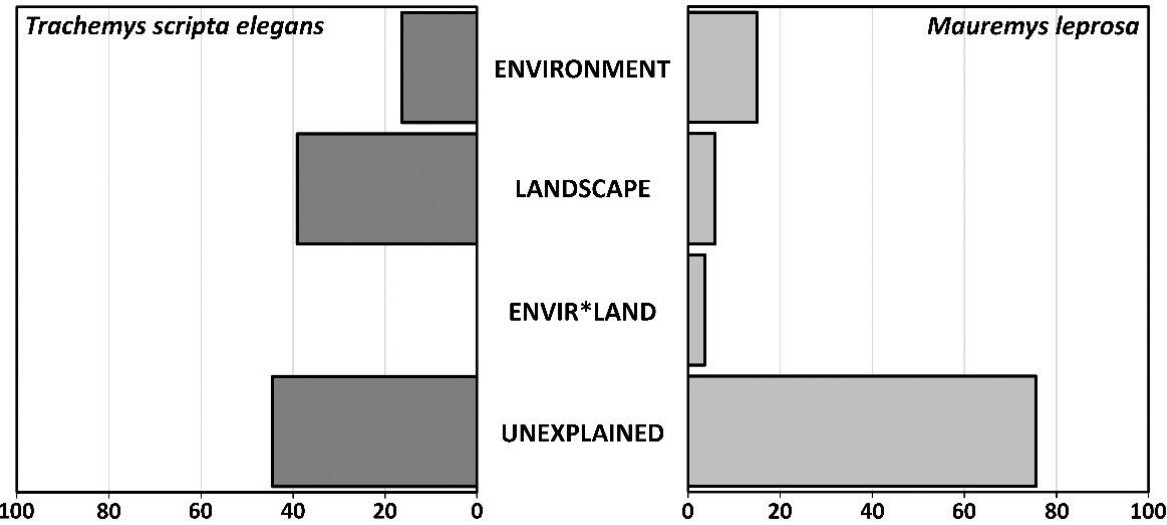

**Figure 3.** Percentage of variance in the distribution of each species explained by the two sets of explanatory variables (i.e., environmental and landscape variables) according to variance partitioning analysis.

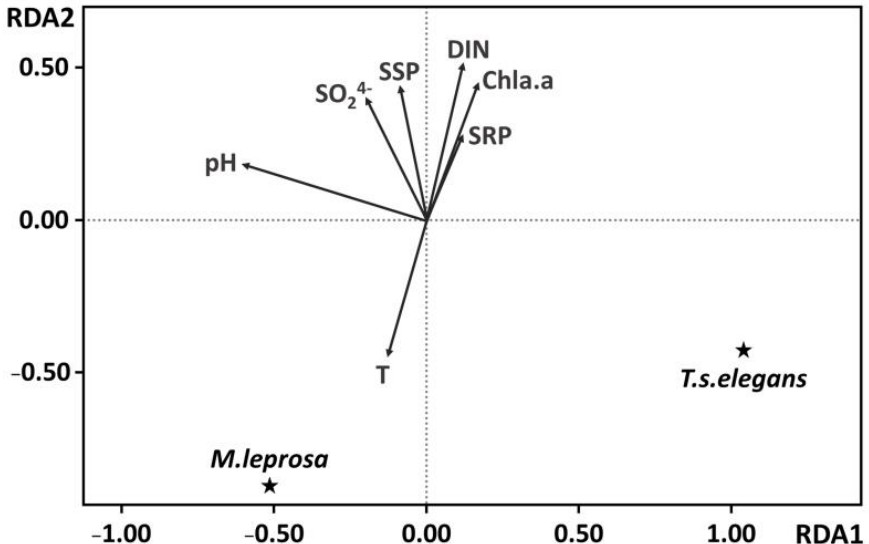

**Figure 4.** Biplot from redundancy analysis showing the relation of the species with the studied environmental variables. Environmental variables are represented by solid vector lines and their acronyms. Species are represented by stars and their name. T: temperature; $SO_2^{4-}$: sulphates concentration; SSP: suspended solids; DIN: dissolved inorganic nitrogen concentration; Chla.a: phytoplanktonic chlorophyll-*a*; SRP: soluble reactive phosphorous concentration.



**Table 4.** Results from latent variable models (LVM). We show two types of correlation. The first (Environmental) is the correlation between the fitted values of the two species [122], representing the correlation between species that can be attributed to a shared/diverging environmental response. The second type of correlation (Residual) was calculated using the latent variable coefficients, also known as factor loadings. It represents the correlation between species that may be attributable to biotic interactions or missing environmental covariates.

|  | Environmental | Residual |
|---|---|---|
| Chl-*a* | −0.34 | −0.43 |
| C⁻ | −0.04 | −0.45 |
| $NH_4^+$ | −0.38 | −0.39 |
| Ox | −0.09 | −0.47 |
| Secchi | −0.12 | −0.53 |
| $Na^+$ | 0.06 | −0.48 |
| SRP | −0.41 | −0.18 |
| SSP | −0.29 | −0.46 |
| T | −0.12 | −0.40 |
| TOC | −0.19 | −0.19 |

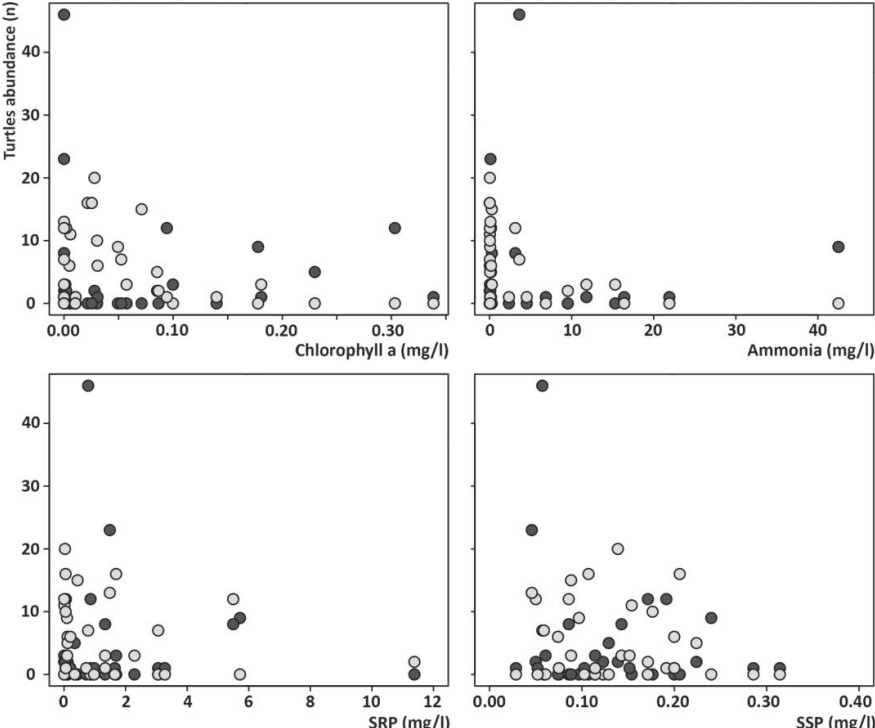

**Figure 5.** Species abundance along the main environmental variables (i.e., those showing statistically significant relationships with the species distribution). Pale gray: *M. leprosa*; dark gray: *T. s. elegans*. SRP: soluble reactive phosphorous concentration; SSP: suspended solids.

The best model explaining the distribution of *M. leprosa* included the following variables: NumP(Ø100) + MPAR(Ø500) + MPFD(Ø500) + MedPS(Ø500) + SEI(Ø2000) + MPAR(Ø2000) + ED(Ø2000) + PSCoV(Ø2000) + PSSD(Ø2000) (Table 5, Figure 6). The model was significant (*p* = 0.017) and explained 33% of the total variance in the species' distribution. None of the landscape variables at 100 m diameter buffer (for proximity or daily movements) were selected by the model. Median patch size (MedPS) and mean perimeter–area ratio (MPAR), both at 500 m diameter buffer (for movements related to the annual cycle of activity), were positively and negatively correlated to *M. leprosa* abundance, respectively. Additionally, the patch-size coefficient of variance (PSCoV) at 2000 m diameter buffer was positively correlated to *M. leprosa* abundance. The best model to explain *T. s. elegans* distribution was SEI(Ø100) + NumP(Ø100) + PSCoV(Ø100) + PSSD(Ø100) +

CA(Ø100) + R(Ø2000) + MedPS(Ø2000) (Table 5) (Figure 6). The model was significant ($p < 0.001$) and explained 67% of the variance in the distribution of the species. At the 100 m diameter buffer (for proximity or daily movements), Shannon's evenness index (SEI), the number of patches (NumP), the patch-size standard deviation (PSSD) and the patch-size coefficient of variance (PSCoV) were negatively correlated to *T. s. elegans* abundance. On the contrary, total core area (CA) showed a positive correlation. The model did not select any landscape variable at a 500 m diameter buffer (i.e., movements related to the annual cycle of activity). Finally, at the 2000 m diameter buffer, richness (R) and median patch size (MedPS) showed a positive and negative correlation with *T. s. elegans* abundance, respectively.

**Table 5.** Statistical values and significance of each variable in the models built for each species using landscape variables at different spatial scales. Buffer diameters related to different movement types of the species: Ø100: proximity movements; Ø500: annual movements; Ø2000: occasional movements. ML: *Mauremys leprosa*; TSE: *Trachemys scripta elegans*. Significance codes: 0 '***'; 0.001 '**'; 0.01 '*'; 0.05 '·'; 0.1 ' '; 1.

| | | Estimate | | Std. Error | | t Value | | Pr (>|t|) | | Sign. | |
|---|---|---|---|---|---|---|---|---|---|---|---|
| | | TSE | ML | TSE | ML | TSE | ML | TSE | ML | TSE | ML |
| | intercept | $-1.901 \times 10^2$ | $2.015 \times 10^2$ | $0.642 \times 10^2$ | $1.222 \times 10^2$ | $-2.963$ | $1.649$ | $0.006$ | $0.112$ | ** | - |
| Ø100 | SEI | $-0.456 \times 10^2$ | - | $6.557 \times 10^0$ | - | $-6.957$ | - | <0.001 | - | *** | - |
| | NumP | $-6.051 \times 10^0$ | $2.604 \times 10^0$ | $1.691 \times 10^0$ | $1.449 \times 10^0$ | $-3.578$ | $1.798$ | $0.001$ | $0.084$ | ** | · |
| | PSCoV | $0.263 \times 10^0$ | - | $0.118 \times 10^0$ | - | $2.221$ | - | $0.035$ | - | * | - |
| | PSSD | $-3.357 \times 10^2$ | - | $0.623 \times 10^2$ | - | $-5.393$ | - | <0.001 | - | *** | - |
| | CA | $3.423 \times 10^2$ | - | $0.868 \times 10^2$ | - | $3.944$ | - | <0.001 | - | *** | - |
| Ø500 | MPAR | - | $-2.306 \times 10^{-3}$ | - | $1.019 \times 10^{-3}$ | - | $-2.263$ | - | $0.033$ | - | * |
| | MPFD | - | $-1.357 \times 10^2$ | - | $7.889 \times 10^1$ | - | $-1.720$ | - | $0.098$ | - | · |
| | MedPS | - | $3.161 \times 10^1$ | - | $1.388 \times 10^1$ | - | $2.277$ | - | $0.032$ | - | * |
| Ø2000 | R | $2.836 \times 10^0$ | - | $1.319 \times 10^0$ | - | $2.151$ | - | $0.041$ | - | * | - |
| | MedPS | $-0.301 \times 10^2$ | - | $0.145 \times 10^2$ | - | $-2.082$ | - | $0.047$ | - | * | - |
| | SEI | - | $5.854 \times 10^1$ | - | $3.176 \times 10^1$ | - | $1.843$ | - | $0.077$ | - | · |
| | MPAR | - | $4.973 \times 10^{-3}$ | - | $2.863 \times 10^{-3}$ | - | $1.737$ | - | $0.095$ | - | · |
| | ED | - | $-1.690 \times 10^{-1}$ | - | $9.427 \times 10^{-2}$ | - | $-1.793$ | - | $0.085$ | - | · |
| | PSCoV | - | $9.766 \times 10^{-2}$ | - | $4.147 \times 10^{-2}$ | - | $2.355$ | - | $0.027$ | - | * |
| | PSSD | - | $-5.398 \times 10^0$ | - | $3.894 \times 10^0$ | - | $-1.386$ | - | $0.178$ | - | - |

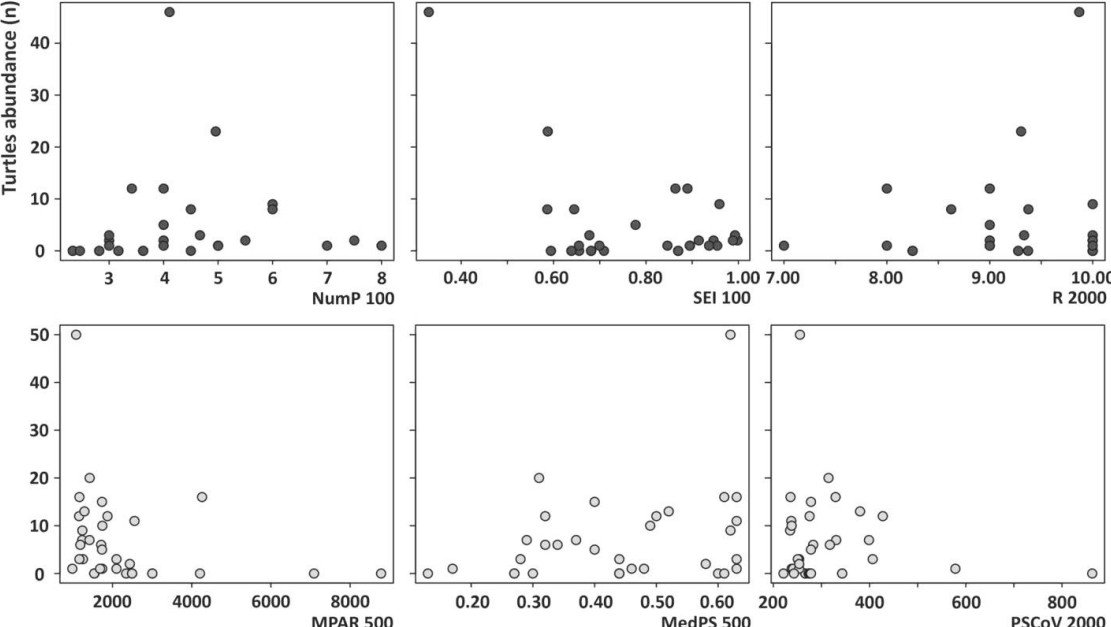

**Figure 6.** Abundance of *M. leprosa* (pale gray) and *T. s. elegans* (dark gray) along the main landscape variables (i.e., those showing statistically significant relationships with the species distribution). NumP: number of patches; SEI: Shannon's evenness index; R: richness; MPAR: mean perimeter–area ratio; MedPS: median patch size; PSCoV: patch size coefficient of variance.

## 4. Discussion

We found that co-occurrence in less than 50% of the sites, and 48% of the variation in their co-occurrence was explained by their biotic interaction or by variables that we did not measure (which might be few since we included a long list of local and landscape variables in our study). Thus, according to previous studies [93–95,102], our results suggest that *M. leprosa* might be displaced by *T. s. elegans*. This could be partly explained by a combination of differences in environmental and habitat preferences and competitive exclusion.

Aquatic ecosystems with high levels of nutrients and organic content (and consequent eutrophication) are rich in trophic resources. This may represent an advantage for generalist species of freshwater turtles [129–131]. The two species in our study seem to be very tolerant of eutrophication. *M. leprosa* has been reported to tolerate eutrophic waters [132,133] and has been found in highly polluted waters [134]. In addition, its sister species, *Mauremys rivulata,* can have very dense populations in eutrophic wetlands [135]. At the same time, *T. s. elegans* has been found in polluted environments within its natural [130] and introduced ranges [131,136]. Regarding salinity, both species appear to tolerate a certain degree of salinity: *M. leprosa* has been recorded in brackish estuarine waters in Portugal [137] and in coastal brackish lagoons along the Mediterranean coast of the Iberian Peninsula [102]. *T. s. elegans* has been found in brackish lagoons (<10 ppm) in South Carolina, USA [138], and in environments with salinities ranging between 0.1‰ and 26‰ in China [139]. In agreement with these results, we did not find strong differences in the environmental preferences of both species. However, the latent variable models showed some moderate negative correlations for SRP, $NH_4$, Chl-*a* and SSP, which aligned with the RDA results. Overall, *M. leprosa* preferred more saline and less eutrophic waters, whereas *T. s. elegans* preferred eutrophic freshwaters. Concordantly, and in the same study area, Franch et al. [102] reported *M. leprosa* from high salinity environments due to the presence of *T. s. elegans* in other environments with lower salinity.

Landscape structure is characterized by the proportion of available habitat, the overall habitat's diversity and the size and arrangement of these in the landscape [140,141]. The study area has been heavily transformed by human activities, causing severe habitat fragmentation and degradation affecting the landscape structure [142]. According to the results, both species are likely to be affected by this landscape transformation at the three scales studied (Ø100, Ø500 and Ø1000) and, therefore, in movement types associated with different stages of their life cycles. The native *M. leprosa* was the least affected by landscape structure of the two studied species. At medium scale, it was positively related with large patches (i.e., high MedPS) of high perimeter values or ecotone area (i.e., high MPAR). A large scale was benefited from a marked heterogeneity in patch sizes (i.e., high PSCoV), whereas *T. s. elegans* abundance was negatively related to the heterogeneity and fragmentation of the surrounding landscape structure. This invasive species prefers uniform landscape within the buffer where daily movements occur, i.e., it has a preference for a low number of different patches, Shannon evenness and patch size heterogeneity. Large-scale landscape structure, related to occasional or sporadic movements of the species, had a weak influence on the distribution of *T. s. elegans*. At this scale, only habitat richness was positively related with *T. s. elegans* abundance, whereas the average patch sizes showed a negative relation. Thus, it seems that at large scales, *T. s. elegans* is associated with a high diversity of habitats of small and uniform relative sizes. These results are consistent with those of Rizkalla and Swihart [143], who showed that *T. s. elegans* was negatively affected by land-use diversity surrounding the wetland.

Freshwater turtles are particularly vulnerable to fragmentation and its consequences (i.e., increased predation pressure or isolation) because of their life history (i.e., long juvenile period, limited fecundity, and dependence on high survival rates of adults) [144]. Wetlands are critical for spawning, hibernation and aestivation and terrestrial dispersal of these animals, and they provide a permanent habitat. While habitat preferences vary from one species to another, all are dependent on land connections between neighboring wetlands [145–147]. In the case of the Llobregat Delta, aquatic habitats might play a key

role in inter/intra-population connectivity, since they are present along the landscape in very different forms (e.g., irrigation channels, lagoons, ponds wetlands) and they provide simple and secure dispersal routes, although water quality tends to be very poor [148,149]. If the study area is framed within a regional scale, beyond the geomorphological deltaic unit, the terrestrial matrix can be expected to have a greater importance as connector with the surrounding landscape. At this scale, landscape structure had an opposite effect on the two studied species.

Thus, the current transformation process of the Llobregat Delta (which is homogenizing of the landscape and decreasing habitat connectivity) could promote the geographical expansion of *T. s. elegans* through its population settlement and consolidation, competition with native turtles, transfers of parasites and diseases, structuring impact on habitats, etc. Despite the high niche breadth and habitat tolerance of *M. leprosa* [60,62,150,151], this species could be severely affected in an indirect way by the Delta transformation process.

Our results suggest that there were some variables with significant influence in the distribution of *M. leprosa* that we did not consider. Since the climatic and environmental conditions of the study area are highly favorable to *M. leprosa*, which has expanded its distribution within the region [152], the unstudied variables may be related to the occurrence of competitive exclusion processes between the native *M. leprosa* and the introduced *T. s. elegans* [153]. Competitive exclusion between the two species has been previously suggested by earlier studies in the same area [102], field observations in Doñana National Park [154], and studies under controlled conditions [91,92]. In addition, previous studies have revealed the existence of different competitive advantages of *T. s. elegans* over *M. leprosa*. For example, *T. s. elegans* tends to monopolize the limited sites appropriate for thermoregulation and displace native turtles to less suitable or suboptimal places [94,155,156]. Less basking can severely affect physiological efficiency (especially digestive) of *M. leprosa* and, consequently, the long-term survival rates of the species [96,153]. Food competition may also play a key role in the co-existence of both species. *M. leprosa* is described as an opportunistic omnivorous species with the ability to modify its diet in response to variability in trophic resources [97], whereas *T. s. elegans* has been described as omnivorous with high carnivorous preferences [106,157,158]. Under controlled conditions, access to food sources for *M. leprosa* is severely restricted by *T. s. elegans*. This species has a dominant aggressive behavior that can seriously affect feeding efficiency of *M. leprosa*, negatively impacting on their survival or reproduction [95]. Another aspect to consider is the chemosensory responses to the presence of freshwater turtles in aquatic habitats. It has been reported that *M. leprosa* prefers aquatic environments with conspecific chemical traces, avoiding those containing traces of *T. s. elegans* [92].

Within the context of habitat generalists species [68,150,152,159], the presence of *M. leprosa* in highly saline and less eutrophic environments and the high unexplained variation in its distribution suggest that its distribution is strongly conditioned by the presence of the invasive *T. s. elegans.* These results may have implications for the conservation of the Mediterranean Pond Turtle. For example, the management of introduced sliders by removing individuals from areas with less salinity may allow *M. leprosa* to recolonize areas where it has been displaced. In addition, restoring natural habitats and promoting habitat heterogeneity might benefit *M. leprosa*.

**Supplementary Materials:** The following supporting information can be downloaded at: https://www.mdpi.com/article/10.3390/land11091582/s1, Table S1: Landscape metrics for three different diameter buffers and for each species; Table S2. Mean values and standard deviations of the environmental variables for each sampling station.

**Author Contributions:** Conceptualization, M.F. (equal), M.C.-A. (equal) and M.R. (supporting); investigation and formal analysis, M.F. (equal) and M.C.-A. (equal); writing—original draft, M.F. (equal), M.C.-A. (equal) and A.M. (supporting); writing—review and editing, M.F. (equal), M.C.-A. (equal), G.A.L. (equal) and A.M. (equal); project administration (FBG302577 project), G.A.L.; funding

acquisition (FBG302577 project), G.A.L. (equal) and A.M. (equal). All authors have read and agreed to the published version of the manuscript.

**Funding:** Data compilation was supported by DMA and Fundació Bosch i Gimpera (FBG302577) 2004–2007. Miguel Cañedo-Argüelles received funding from the People Program (Marie Curie Actions) of the Seventh Framework Program of the European Union (FP7/2007–2013) under grant agreement no. 600388 of REA (TECNIOspring Program), the Agency for Competitiveness and Business of the Government of Catalonia (ACCIÓ) and was supported by a Ramón y Cajal contract funded by the Spanish Ministry of Science and Innovation (RYC2020-029829-I). The formal analysis and the manuscript writing process have been self-financed by the authors.

**Institutional Review Board Statement:** The animal study protocol was approved by the Institutional Review Board (or Ethics Committee) of Environmental Department of the Catalan government (DMA) (capture permits SF/250 for 2004; SF/227 for 2005) for studies involving capture and release of wild animals.

**Informed Consent Statement:** Not applicable.

**Data Availability Statement:** Not applicable.

**Acknowledgments:** We thank all collaborators of the Herpetology group of Universitat de Barcelona and four anonymous reviewers for their helpful comments and suggestions that greatly improved the last version of the manuscript.

**Conflicts of Interest:** The authors declare no conflict of interest.

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
