# Peer review of "Coexistence of Native and Invasive Freshwater Turtles: The Llobregat Delta (NE Iberian Peninsula) as a Case Study"

_land, doi:10.3390/land11091582_

Round 1

Reviewer 1 Report

  The elaboration of the article is very careful.
The methodology developed is sufficiently clear and leaves no doubt as to how the experiment was developed.
The results section is well elaborated, supported by the methodology, as well as the discussion of the results.

For all these reasons, it seems to me that this article is ready to be published. Congratulations for the work.

Author Response

Response to Reviewer 1 Commments:

Thank you very much for your comments on the manuscript.

This was part of a study carried out a long time ago and that was part of the first author's doctoral thesis, although it was rejected in a journal. Publishing it some time later has entailed a great effort on the part of all the authors.

Again, thank you very much for the very lucky and encouraging comments.

Reviewer 2 Report

I have read the manuscript entitled “Coexistence of native and invasive freshwater turtles: the Llobregat Delta (NE Iberian Peninsula) as a case studyby Marc Franch and colleagues, submitted to the Land magazine. I think, it is interesting and well written manuscript on problems on a coexistence of native and alien species of turtles, i.e. the Mediterranean stripe-necked turtle Mauremys leprosa and the Red-eared Slider Trachemys scripta elegans.

I have some specific comments and questions, which could help to improve the manuscript.

Specific comments and questions:
line 25 “The species showed a very low niche overlap”. I believe it is true, but it could be effect (see e.g. line 377) “of competitive exclusion processes between the native M. leprosa and the introduced”.  I think it should be mentioned in the Abstract section.

Figure 1. The figure could be improved, I think. For example, it is difficult to find “The black dots represent our sampling stations”. The location of the study area (i.e., the square on top left) will useful for persons knowing map of Europe very well; for other persons, the small map will be not enough, I think.

line 118. What is means “p.e.”? It is not typical abbreviation, thus should be explained.

lines 130-131: “all captured turtles were marked, measured”. How the turtles were marked? It could be important information for the one-year-long study.
What about sizes of them? – even general information on sizes of the captured turtles would be useful for other scientists.

lines 177-178: “having a Spearman correlation coefficient higher than 0.70 or lower than -0.70 [116,117] was removed”. What about statistical significance of the correlation coefficients? Even general information in the subject would be useful for readers.

line 217 “statistical computing software R”. I think, that the used version of the software should be mentioned here.

line 220-222: “The two species showed a co-occurrence of 46.15% in the studied water bodies, 38.46% of the water bodies only had T. s. elegans and 15.38% with only M. leprosa (Table 3, Figure 2).”
What about the water bodies in which one turtle was captured, only? If data gathered on such localities were used in the analyses? If you remove data from such localities – is the conclusion similar? I believe that yes, but it could be interesting and important for readers.
Additionally: do you have data on the turtle occurrence based on direct observations (e.g. basking animals)? If the data based just on capturing animals is reliable (to analyse co-occurrence of the species, if in some water bodies, one or two animals were captured, only)?

Table 4. What about statistical significance of the correlation coefficients? Even general information would be useful for readers.

lines 342-343 “This invasive species prefers uniform landscape within the buffer where daily movements occur”. If the distribution could be effect of the places of release the Red-eared Sliders by people?

Important parameter is availability of food, but it was not studied, I think. Do you have (even general) data on food availability in the sampling stations? More information on the subject in the discussion section is recommended.

I think it will be better use the word “turtle” for Mauremys leprosa. Now, in the lines 19 and 400 the word “terrapin” is used for the species. It could be confusing for some readers.

Author Response

Thank you very much for your comments on the manuscript. Please see the attachment.
